# Randomised controlled trial to establish the clinical and cost-effectiveness of expectant management versus preoperative imaging with magnetic resonance cholangiopancreatography in patients with symptomatic gallbladder disease undergoing laparoscopic cholecystectomy at low or moderate risk of common bile duct stones (The Sunflower Study): a study protocol

For numbered affiliations see end of article.

**Correspondence to**
Madeleine Clout;
madeleine.clout@bristol.ac.uk

Madeleine Clout ,[1] Jane Blazeby ,[2] Chris Rogers,[1] Barnaby Reeves ,[1] Michelle Lazaroo,[1] Kerry Avery ,[2] Natalie S Blencowe ,[2] Ravi Vohra,[3] Neil Jennings,[4] William Hollingworth,[5] Joanna Thorn ,[5] Marcus Jepson,[5] Jane Collingwood,[5] Ashley Guthrie,[6] Elizabeth Booth,[7] Samir Pathak,[8] Ian Beckingham,[9] Lucy Culliford,[1] Ewen A Griffiths,[10] Raneem Albazaz,[6] Giles Toogood,[11] On behalf of the Sunflower Study Executive Group

## ABSTRACT

**Introduction** Surgery to remove the gallbladder (laparoscopic cholecystectomy (LC)) is the standard treatment for symptomatic gallbladder disease. One potential complication of gallbladder disease is that gallstones can pass into the common bile duct (CBD) where they may remain dormant, pass spontaneously into the bowel or cause problems such as obstructive jaundice or pancreatitis. Patients requiring LC are assessed preoperatively for their risk of CBD stones using liver function tests and imaging. If the risk is high, guidelines recommend further investigation and treatment. Further investigation of patients at low or moderate risk of CBD stones is not standardised, and the practice of imaging the CBD using magnetic resonance cholangiopancreatography (MRCP) in these patients varies across the UK. The consequences of these decisions may lead to overtreatment or undertreatment of patients.

**Methods and analysis** We are conducting a UK multicentre, pragmatic, open, randomised controlled trial with internal pilot phase to compare the effectiveness and cost-effectiveness of preoperative imaging with MRCP versus expectant management (ie, no preoperative imaging) in adult patients with symptomatic gallbladder disease undergoing urgent or elective LC who are at low or moderate risk of CBD stones. We aim to recruit 13 680 patients over 48 months. The primary outcome is any

## Strengths and limitations of this study

► This is a randomised controlled trial with randomisation taking place using a secure internet-based system to ensure allocation concealment, which will ensure that patients allocated to each group have comparable baseline characteristics.

► The study includes an internal pilot phase with progression to the main study dependent on a number of criteria to ensure that the study will only proceed if it is viable to do so.

► Participants, clinicians and other hospital staff caring for participants will not be blind to the study allocation.

► The criteria used to define low and moderate risk of common bile duct stones have been identified by clinicians within the study team, as there is no universal agreed definition of these terms.

► The study primary outcome will be derived from routine data sources, which should ensure high levels of data completeness but is reliant on timely approval of numerous applications to holders of routine data.

hospital admission within 18 months of randomisation for a complication of gallstones. This includes complications of endoscopic retrograde cholangiopancreatography for the

treatment of gallstones and complications of LC. This will be determined using routine data sources, for example, National Health Service Digital Hospital Episode Statistics for participants in England. Secondary outcomes include cost-effectiveness and patient-reported quality of life, with participants followed up for a median of 18 months.

**Ethics and dissemination** This study received approval from Yorkshire & The Humber – South Yorkshire Research Ethics Committee. Results will be submitted for publication in a peer-reviewed journal.

**Trial registration number** ISRCTN10378861.

## INTRODUCTION

Surgery to remove the gallbladder, known as laparoscopic cholecystectomy (LC), is the standard treatment for symptomatic gallbladder disease. LC is one of the most common operations undertaken globally, and around 70 000 LC operations are performed annually in England.[1]

One potential complication of gallbladder disease is that gallstones may pass from the gallbladder into the common bile duct (CBD). Once in the CBD, they may remain dormant, pass spontaneously into the bowel or cause problems such as obstructive jaundice, cholangitis or pancreatitis. If CBD stones are found, current guidelines recommend removal either before or during LC. Therefore, patients requiring LC are assessed preoperatively for their risk of CBD stones using a combination of liver function tests (LFTs) and imaging. If the risk is high, guidelines recommend further investigation and treatment.[2–6] Further investigation of patients at moderate or low risk of CBD stones is not standardised, and practice across the UK varies. Fewer than 10% of these patients actually have CBD stones.[2–4 6 7] Consequently, surgeons make different decisions about investigation of the CBD; some surgeons choose to perform diagnostic imaging tests for these patients and others do not. This variation likely results in overtreatment or undertreatment, with significant risks to the patient in terms of morbidity, health-related quality of life (HRQoL) and costs to the health service.

A UK-wide audit found that a third of patients undergoing LC, irrespective of risk of CBD stones, had diagnostic imaging to test for CBD stones.[7] This is most frequently performed with magnetic resonance cholangiopancreatography (MRCP) before admission for LC; however, some surgeons instead use intraoperative cholangiogram and other intraoperative ultrasound examinations of the CBD. MRCP produces detailed images of the biliary and pancreatic ducts, involves a 1-hour hospital visit and costs the National Health Service (NHS) about £365. MRCP can identify CBD stones but may delay LC, which can have a detrimental effect on HRQoL and can lead to increased problems with gallstones or a more complex LC.[8] If a CBD stone is identified during MRCP, extraction is most frequently performed before LC with an endoscopic retrograde cholangiopancreatography (ERCP). The risks and mortality associated with ERCP are significant,[9] as is the cost to the NHS (about £1600). Not having MRCP avoids these risks but can lead to CBD

stones remaining after surgery, which may also cause complications. Research is needed to establish whether it is appropriate for patients at low and moderate risk of CBD stones to proceed straight to LC without imaging the CBD with MRCP.

## METHODS AND ANALYSIS
### Aims and objectives

The Sunflower study aims to compare the effectiveness and cost-effectiveness of preoperative imaging with MRCP versus no MRCP (hereafter called expectant management (EM)) in patients with symptomatic gallbladder disease undergoing LC who are at low or moderate risk of CBD stones. The study will test the hypothesis that EM is not inferior to MRCP with respect to hospitalisation for treatment for a complication of gallstones up to 18 months after randomisation.

The objectives are to estimate the:
1. Difference between groups in the proportion of participants requiring a hospital admission for treatment of a complication of gallstones in the gallbladder or CBD.
2. Difference between groups with respect to a range of secondary outcomes, including patient-reported HRQoL, symptoms related to complications of gallstones in the gallbladder or CBD and symptoms related to complications associated with LC and, where applicable, ERCP.
3. Cost-effectiveness of MRCP compared with EM.

### Study design

The Sunflower Study is a multicentre, pragmatic, open, randomised controlled trial with an internal pilot. The study has an embedded Qualitative Research Integrated within Trials Recruitment Intervention (QRI) to optimise informed consent and recruitment.[10]

There are two phases to the study:

Phase 1 (internal pilot): set-up and recruit from at least 36 centres by month 16 of recruitment, with integrated QRI to optimise recruitment.

Phase 2: increase the number of centres to at least 50 and continue recruitment using the methods established in phase 1, along with integrated QRI to optimise recruitment and promote adherence in new centres.

Progression to phase 2 is conditional on the study demonstrating to the funder that: (A) at least 30 centres are open and recruiting; (B) at least 1750 participants have been randomised; (C) at least 90% of participants have followed their allocated pathway; and (D) the primary outcome can be identified reliably from routine data.

### Setting

Recruitment will take place at a minimum of 50 NHS hospitals across the UK with a recruitment target of 13 680 participants over a 48-month period (figure 1).

### Study population

The target population is adults (18 years or older) with symptomatic gallbladder disease, scheduled and fit for

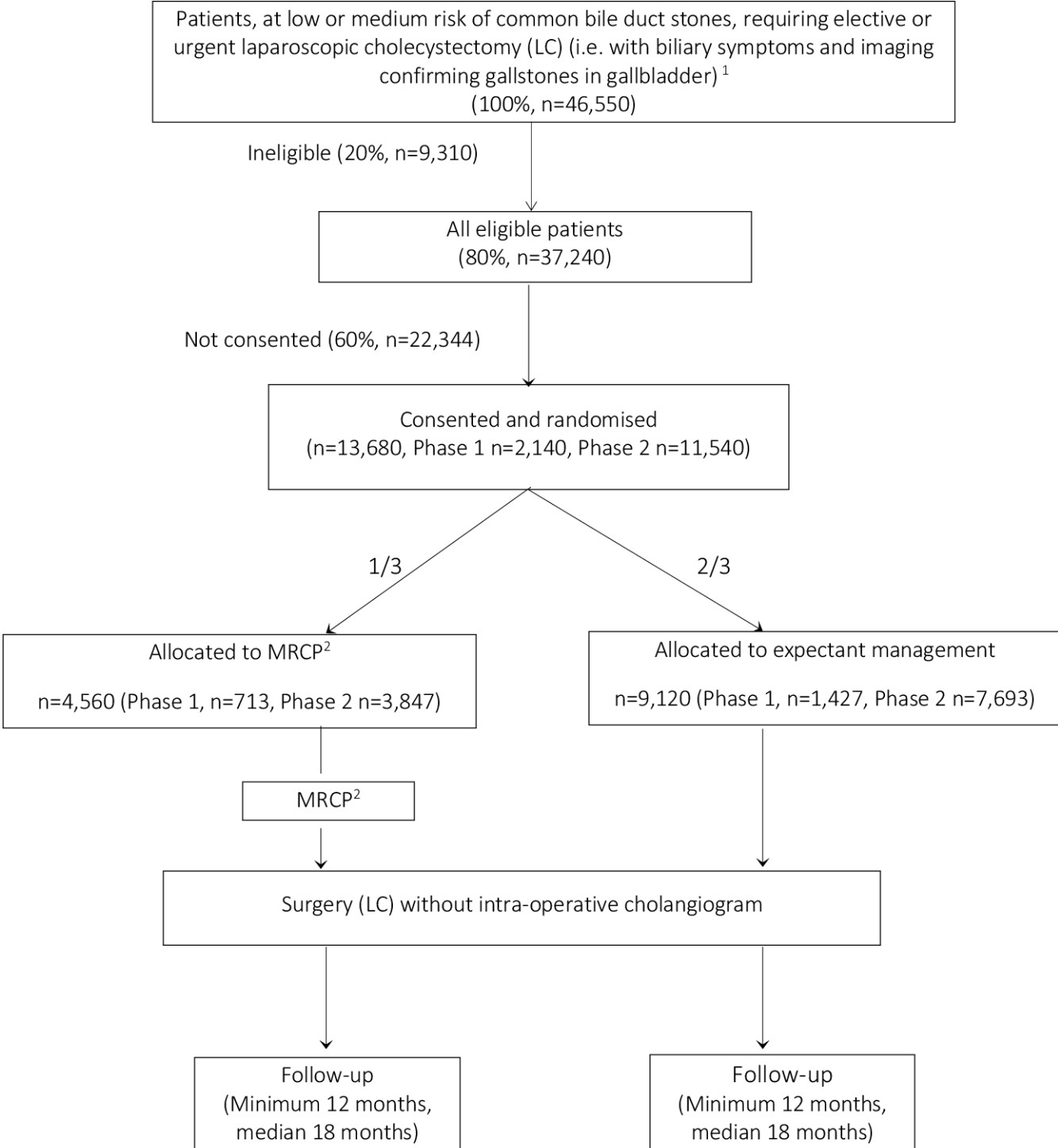

**Figure 1** The study schema. MRCP, magnetic resonance cholangiopancreatography.

LC as an elective or urgent procedure who are at low or moderate risk of CBD stones. Low or moderate risk is defined as CBD diameter ≤8 mm on ultrasound scan (USS), bilirubin ≤50 µmol/L and alanine transferase (ALT) less than three times the upper limit of normal (≤3×ULN) and/or alkaline phosphatase (ALP) ≤3×ULN. If the CBD cannot be visualised on USS, the patient may be recruited if no intrahepatic duct dilatation is reported.

Patients are excluded if they are unable to undergo MRCP, have evidence of empyema or perforated gallbladder requiring urgent intervention, have had a previous gastric bypass, have had a previous MRCP or endoscopic USS within the last 3 months, have had any previous ERCP or have haemolytic disease. Patients must be willing to participate in follow-up and be able to provide informed consent.

## Primary outcome

The primary outcome is any of the following:

i.    Any hospital admission within 18 months of randomisation for treatment of a complication of gallstones whether in the CBD or gallbladder.

ii.   Complications during the admission for LC for the treatment for gallstones or any readmission for complications of the LC leading to a hospital day of >2 days. Complications will include, but not be limited to, return to theatre post-LC for any cause, percutaneous radiological drainage and ERCP for non-diagnostic reasons (eg, for a bile leak). It does not include a diagnostic ERCP performed following an MRCP where CBD stones were identified.

iii.  Complications during any ERCP for the treatment for gallstones. Complications will include, but are not limited to, blood transfusion post-ERCP, percutaneous radiological drainage, treatment of a perforation occurring during ERCP, acute pancreatitis and other complications leading to a hospital stay of >2 days.

This outcome will be ascertained by linking the study data with routinely collected Hospital Episode Statistics (HES) and mortality data from NHS Digital for participants recruited in England and the equivalent data sources for participants recruited in Scotland, Wales and Northern Ireland. Routine data sources contain information on care provided to all patients treated in UK NHS hospitals.[11] The final specification of qualifying events for identifying the primary outcome from these routine sources will be developed and validated during phase 1 of the study, through comparison of routine data with events identified from review of the medical records at 90 days post-LC.

## Secondary outcomes

Patient-reported HRQoL will be measured using the EQ-5D-5L questionnaire.[12] To minimise the overall study burden, about 20% of participants will be sampled and asked to complete questionnaires at multiple time points (table 1). A questionnaire pack will be provided to participants containing the validated EQ-5D-5L questionnaire plus questions about the impact of abdominal pain on work, productivity and primary healthcare use.

Other secondary outcomes include items that cover the domains in the core outcome set for patients with symptomatic gallstones and hospital resource use in the 18 months after randomisation collected from routine data sources.[13 14]

## Sample size

The sample size of 13 680 has been chosen to test the hypothesis that EM is non-inferior to MRCP with respect to the primary outcome. In estimating the sample size, we have considered the proportion of patients that would be expected to experience the primary outcome, as identified in the CholeS audit (5%–10%) and through exploration of HES data.[7] This includes patients at high risk of CBD stones. The consensus among clinicians on the study

team was that the non-inferiority margin should be set at 1.5%, that is, that the risk of the primary outcome with EM should not exceed 8.5% assuming a risk of 7% after MRCP. A study of 13 680 participants will provide 90% power to test the non-inferiority hypothesis for a 7% event risk with MRCP and 80% power to test the non-inferiority hypothesis for a higher event risk of 8.5%, assuming 2.5% one-sided statistical significance and a 1:2 allocation ratio (MRCP:EM). The 20% sample of participants providing HRQoL data will be stratified by centre. This sample of 2736 participants will have >90% power to detect a small effect size of 0.12 standard deviation between groups on the EQ-5D-5L questionnaire.

## Recruitment

Recruiting centres will screen LC waiting lists and gallbladder clinic lists for potentially eligible patients. Potential participants will also be identified from urgent inpatient and ambulatory care admissions and ward handovers. Eligible patients will be provided with study information and will have the opportunity to ask questions before being approached for their informed consent (see online supplemental material 1). Patients who feel they have had insufficient time to consider the study will be invited to take home a consent form and questionnaire pack (where applicable) and to complete and return them by post. Details of all patients approached for the study and reasons for non-participation will be documented.

## Randomisation

Randomisation will be performed by an authorised member of the local research team using a secure internet-based system to ensure allocation concealment. The allocation will be computer generated and stratified by centre.

Participants will be randomised to MRCP or EM in a 1:2 ratio. This ratio has been selected to reflect current levels of MRCP provision (13%–26% of patients at low or moderate risk of CBD stones currently undergo MRCP).[7]

## Interventions

The study interventions are preoperative MRCP and EM.

Participants allocated to the MRCP group will have an MRCP before the date of their listed LC. The study will make no changes to the usual hospital radiology protocols used for MRCP and will not impose any timelines for the MRCP or any subsequent interventions. If a CBD stone is found on MRCP, surgeons can manage extraction according to their usual practice. The study does not stipulate removal of identified CBD stones, and any further interventions are at the discretion of the clinician. If an ERCP is requested, it will be performed as a separate inpatient procedure before admission for LC.

Participants allocated to EM will be listed for LC without any preoperative imaging. If MRCP is carried out, for example, due to a change in a participant's clinical

**Table 1** Data collection time points

| Data item | Prerandomisation | Presurgery | Hospital admission for LC | 90 days post-LC | 3 months* | 6 months* | 12 months* | 18 months* |
|---|---|---|---|---|---|---|---|---|
| Eligibility | ✓ | | | | | | | |
| Written informed consent | ✓ | | | | | | | |
| Medical history | ✓ | | | | | | | |
| EQ-5D-5L, productivity and primary care use questionnaire (20% sample) | ✓ | | ✓‡ | | ✓ | ✓ | ✓ | ✓ |
| MRCP, ERCP and intraoperative imaging details, if applicable | | ✓ | | | | | | |
| Operative and postoperative details | | | ✓ | | | | | |
| Items in the symptomatic gallstone core outcome set | | | | ✓ | | | | |
| Safety data | | | | ✓ | | | | |
| Study consultations audio recorded for QRI† | ✓ | | | | | | | |

*These time points are months post randomisation.
†During phase 1 of the study, some recruitment consultations will be audio-recorded at high volume centres with patient consent. Thereafter, some consultations will be recorded at centres where recruitment rates fall below target.
‡Participants will not be asked to complete the LC admission questionnaires if they have completed the baseline questionnaires within the previous 2 days.
ERCP, endoscopic retrograde cholangiopancreatography; LC, laparoscopic cholecystectomy; MRCP, magnetic resonance cholangiopancreatography; QRI, Qualitative Research Integrated within Trials Recruitment Intervention.

circumstances after randomisation, this will be documented as a crossover.

In both groups, intraoperative imaging will not be permitted unless there is an anatomical reason to do so, or if a CBD stone is found on MRCP. Any deviations from the protocol will be recorded and monitored.

### Data collection

Data will be collected by the local study team at baseline, before, during and after LC and at hospital discharge (table 1). Data collection will be from hospital records, including operative notes, and will include information on any pre-LC MRCP and ERCP and any intraoperative imaging. Data will either be collected onto a paper case report form and transcribed into a secure electronic study database or entered directly into the database. During phase 1, safety data will also be collected from hospital records 90 days post-LC by the local study team.

### Linkage to routine data

With consent, participant identifiers will be sent to NHS Digital and equivalent organisations for participants recruited outside of England. These organisations will match and extract pseudonymised data for these participants for all episodes of hospital care for a minimum of 18 months from the point of randomisation and for a period of 12 months prior to randomisation. The data will include inpatient and critical care episodes, accident and emergency department attendances, outpatient appointments, diagnostic imaging investigations and mortality.

### MRCP quality assurance

To address the challenge of MRCP technique varying across participating centres, images and reports for a random 10% sample of participants who receive an MRCP will be transferred for independent quality assurance review by consultant radiologists. The MRCP examinations will be assessed for the type of sequences performed, the diagnostic quality of the images and level of concordance with the hospital reports. These data will provide an overview of current NHS practice and the overall quality of MRCP examinations, as well as characterising variation in MRCP technique in study centres.

### Statistical analysis

The study will be analysed on an intention-to-treat (ITT) basis, that is, outcomes will be analysed according to the allocated treatment pathway irrespective of future management and events, and every effort will be made to include all randomised participants. Non-adherence to the treatment allocation will be documented. The primary analysis will follow the Consolidated Standards of Reporting Trials guidelines for a non-inferiority study.[15] An analysis according to treatment pathway followed will also be performed for the primary outcome. As recommended, both analyses will be considered when assessing whether the hypothesis is met.[16] The primary outcome will be compared using survival methods to allow for censoring. For participants without a qualifying primary

outcome event, they will be censored at the time the dataset was compiled. If more than one qualifying event occurs (eg, two hospital admissions for gallstone-related complications), time to the first event will be used. The frequency of each element of the primary outcome and reasons for admission, where applicable, will be described. Secondary outcomes will be compared using a mixed model for continuous outcomes or a generalised linear regression model for binary outcomes, adjusted for prerandomisation measures when available. For secondary outcomes measured at multiple time points, changes in treatment effect with time since randomisation will be assessed by adding a treatment by time interaction to the model and comparing models using a likelihood ratio test. Model fit will be assessed, and alternative models and/or transformations (eg, to induce normality) will be explored where appropriate. Sensitivity analyses using multiple imputation for missing data will be explored. Analyses will be adjusted for centre, and treatment differences will be reported with 95% confidence intervals.

A detailed analysis plan will be prepared. There is no intention to compare any outcomes between groups at the end of phase 1; only descriptive statistics about eligibility, recruitment and adherence will be summarised at this time to determine whether the study satisfies the progression criteria.

Exploratory analyses will include:
i. Relationship between number and size of stones seen on MRCP and patient outcome (cohort undergoing MRCP only).
ii. Relationship between stones removed under ERCP or not (eg, 'necessary' vs 'unnecessary' ERCP) and patient outcomes (cohort undergoing ERCP only).

Subgroup analyses will evaluate the primary outcome in patients defined by characteristics. The main subgroup analysis will be in patients with low versus moderate risk of CBD stones, which will be defined following phase 1. This definition is expected to be a composite of baseline LFTs, baseline CBD diameter on USS and LC admission type (elective or urgent). Further subgroups will be defined:
i. Patients referred for elective surgery versus patients undergoing urgent surgery.
ii. Patients with normal LFTs at baseline (ie, low risk) versus patients with abnormal (outside of local Trust upper and/or lower normal limits) LFTs at baseline (ie, moderate risk).
iii. Patients with a history of pancreatitis versus patients with no history of pancreatitis.

### Health economic analysis

The primary economic evaluation will compare NHS costs and patient outcomes, measured by quality-adjusted life years (QALYs), between the groups on an ITT basis. This analysis will explore whether the initial savings due to not using MRCP are offset by higher treatment costs and worse patient outcomes due to LC complications and/or retained stones.

English NHS tariffs will be used to estimate the costs.[17] These tariffs are based on Healthcare Resource Groups (HRGs) and distinguish between open/laparoscopic, elective/urgent and day case/inpatient surgery. However, they are not sufficiently granular to measure the impact on costs of small differences in theatre time or postsurgical length of stay, which might be evident between groups. Therefore, micro-costing may also be used to estimate the incremental costs of LC. HRG codes, NHS tariffs and national unit costs will be used to estimate all other care costs during follow-up.[18] National Institute for Health and Care Excellence recommended EQ-5D-5L value sets for England will be used with linear interpolation between time points and adjustment for baseline to calculate QALYs.[19 20]

Costs and outcomes beyond 12 months will be discounted at standard rates.[21] Missing cost and EQ-5D-5L data will be described, and multiple imputation techniques will be used as appropriate. The incremental cost per QALY of MRCP versus EM over the 18-month follow-up period will be estimated, and bootstrapping techniques will be used to estimate 95% CIs. Regression analyses will be used to estimate the incremental net monetary benefit and cost-effectiveness acceptability curve of MRCP at conventional thresholds after controlling for key baseline covariates.[22] In a secondary economic analysis, we will estimate the cost per gallstone-related readmission avoided. Sensitivity analyses will explore the robustness of the conclusions to plausible differences in key costing assumptions. If there is evidence that costs and outcome differences between groups persist between 6 and 18 months, a simple extrapolation model to estimate cost-effectiveness beyond 18 months will be considered. The impact of care pathways on patient productivity costs will be described in secondary analyses. A health economics analysis plan will detail analytical methods.

## Safety

In accordance with Good Clinical Practice guidelines, the occurrence of adverse events will be monitored carefully and recorded in detail.

Following LC, transient postoperative complications are not unexpected and often delay the patient's hospital discharge. These complications are classified as expected events in this study. There are also some known complications of ERCP and MRCP, which are also classified as expected events. Any event classified as expected will not require expedited reporting to the sponsor or research ethics committee (REC), unless in the event of a participant death. Unexpected serious adverse events, including participant deaths, will be reported to the study sponsor and coordinating centre.

Data on adverse events will be collected from randomisation until hospital discharge after LC, or for 9 months postrandomisation for participants who do not undergo LC for any reason.

## Patient and public involvement statement

We will work in close partnership with patients and members of the public throughout all phases of the Sunflower Study. The study funding proposal was developed in collaboration with our patient/public contributor coapplicant who has experience of gallstone investigations/surgery. Two public contributors have been appointed to the study steering committee (SSC), who participate in all committee meetings.

Feedback from patient and public contributors on study rationale and design was sought during the prefunding and postfunding stages of the study. Topics discussed included experiences of gallstone investigations and bile duct imaging, advantages/disadvantages of bile duct imaging, relevance/importance of a randomised trial and barriers/facilitators to recruitment, randomisation and follow-up to inform development of the study protocol.

A patient/public advisory group has been established, and patient and public involvement activities have been conducted regularly throughout the set-up and running of the study. Consultations with patient/public contributors have covered a diverse range of issues. For example, patient/public contributors have been involved in drafting and revising study participant information leaflets and consent forms to improve participant information provision. We have also consulted with patient/public contributors regarding proposed protocol amendments, resulting in the introduction of an additional postal/remote consent pathway to optimise recruitment during the COVID-19 pandemic. Interviews with patient/public contributors have also been conducted to explore concerns around study participation and attending hospital during the pandemic. Most participants were in favour of recruitment restarting, helping to inform subsequent decisions around restarting the study. Further patient/public activities have been undertaken to explore additional strategies for optimising recruitment (including the codevelopment of a patient-facing clinic recruitment poster), patients' experiences/burden of study participation, and layout and content of the Sunflower Study website.

We will work closely with patient and public contributors to develop a strategy for sharing the study findings with participants, future patients, service users and members of the wider public. This will include consideration of disseminating findings using a range of materials and methods, such as plain language summaries, newsletters, videos, social media, websites and patient support groups.

## Risk of bias

Participating centres will aim to screen all adult patients with symptomatic gallbladder disease who are scheduled and fit for LC. We expect that the recruited patients will be generalisable to the wider population. Patients allocated to each group will have comparable baseline characteristics as randomisation will take place after consent via a secure website with allocation concealment. Allocations will be stratified by centre to minimise confounding.

The primary outcome is objective and will be obtained from routinely collected data sources. This is important

as participants, clinicians and other hospital staff caring for participants will be unblinded. Attempts to blind participants by undertaking 'dummy' MRCP in those randomised to EM would have added significant additional research costs and created logistical issues. Because 'dummy' MRCP would cause a similar delay to LC as actual MRCP, it would also prevent the study from assessing the impact of this on outcomes.

The risk of missing data has been minimised by using routinely collected data for the primary outcome and by undertaking regular review of missing data captured explicitly for the study.

We do not expect patient-reported outcomes to be at risk of bias, since we do not anticipate that participants will have preferences or expectations about the use of MRCP. We aim to promote uncertainty about the effects of the two interventions by providing information that describes the risks and benefits of each intervention. This information has been reviewed by a public and patient involvement group and informed by the experiences of the QRI researchers.

### Study management and oversight

Preparation of study documents, site initiation visits and training, day-to-day running of the study and monitoring of centres are managed by the Clinical Trials and Evaluation Unit, Bristol Trials Centre. A study executive group is overseeing the study and meets regularly to review milestones. A data monitoring and safety committee meets biannually to review data. An SSC oversees the study and has ultimate responsibility for any decision about its continuation. Membership of the study oversight committees is described in the Acknowledgements section.

### Changes to the study protocol

Two substantial amendments have been approved by the REC. The first, in April 2019, added the option of postal consent, and the second, in December 2019, revised some inclusion/exclusion criteria and clarified information relating to withdrawals in line with the Health Research Authority guidance.

### DISCUSSION

The main challenge facing the Sunflower Study is the recruitment target of 13 680 participants. Recruitment started 1 month later than planned in January 2019. The initial focus was to open centres to recruitment, with a target of 36 centres recruiting within phase 1. The study has already exceeded this target. As of 23 August 2020, 48 centres were open to recruitment, and 2040 participants had been randomised. The QRI will be key to optimising recruitment, providing training at investigators meetings and recruitment tips for staff at participating centres.[23] [24] With a 48-month recruitment period, it will also be key for the study to keep centres engaged. Initiatives include the use of an active Twitter account, regular recruitment updates and centre targeted emails,

biannual newsletters and annual investigators' meetings. It will also be important to remain aware of other relevant research that could impact on the study. The study team will undertake annual literature reviews.

The study includes patients who are scheduled for LC as either an elective or urgent procedure. It is important to ensure that both pathways are represented to promote generalisability of the study results, but it is recognised that recruitment of urgent patients is challenging. These patients can be admitted at any time of day and over weekends when research nurses may not be available to approach patients and receive consent. As of 23 August 2020, 77.8% of randomised participants were elective referrals and 22.2% were from urgent admissions. The study is keen to involve surgical trainees in the recruitment process and is participating in the National Institute for Health Research associate Principal Investigator scheme.[25] During study training, the local team is encouraged to consider recruitment from the urgent pathway, and the ratio of elective to urgent patients recruited is being monitored. The study feels that surgical trainees will be critical to the successful recruitment of urgent patients. Discussions from an investigator meeting identified that the inclusion criterion, ALT and/or ALP less than 2 ×ULN, was disproportionally excluding urgent patients, and so this was amended to 3× ULN. The impact of this amendment will be monitored.

For the study results to be generalisable, we aim to recruit both patients classified as being at low and moderate risk of CBD stones. Low-risk patients are defined as those with bilirubin, ALT and ALP all within the normal range of their local Trust. As of 23 August 2020, 67% of randomised participants were classified as being at low risk and 33% as moderate risk.

Another major challenge will be obtaining routinely collected data from England, Wales, Northern Ireland and Scotland. Applications to obtain routinely collected data are complex and time consuming, and a different application process is required for each nation. The coordinating centre has considerable experience in completing these applications, but they must be successful and timely for the primary outcome to be determined and health economic analyses to be completed.

The primary outcome for the study is composed of three elements. The first will compare hospital admissions for treatment of gallstone complications in the 18 months after randomisation. The second and third elements will compare postoperative complications of LC and complications of ERCP. These two elements are important to measure, as there is evidence of increased rates of some complications (eg, biliary leakage) in patients with retained stones post-LC.[26] This may have relevance to participants randomised to EM. It is also possible that participants are overtreated with MRCP, and any subsequent ERCP, and suffer avoidable complications. These elements will help inform study results and application in practice. The primary outcome will be reported both overall and by each individual element.

## Study status

Recruitment to the study started in January 2019 and is due to complete in January 2023. The current protocol is version 3.0 dated 25 November 2019. Relevant regulatory approvals will be obtained for amendments to the study documentation. Information relating to amendments will be disseminated to relevant parties via email, newsletter and an electronic study management system.

## Ethics and dissemination

This study was reviewed and given a favourable opinion by Yorkshire & The Humber – South Yorkshire REC on the 10 December 2018 (REC reference: 18/YH/0358). Each participating centre is required to provide evidence of local confirmation of capacity and capability prior to starting recruitment to the study. Informed consent to participate will be obtained from all study participants. Participants have the right to withdraw from the study at any time and, if they do withdraw, will be treated according to standard local hospital procedures. Routine data will continue to be collected for withdrawn participants, unless the participant explicitly asks for this data collection to stop.

The study findings will be disseminated through peer-reviewed publications and international conferences and will be provided to study participants. A study specific Twitter account will be used to promote the study, provide updates on milestones and disseminate study findings. After study completion and reporting, the technical appendix, statistical code and dataset generated from the study will be available from the corresponding author on request. The request must include a specification of the data requested and justification for that request (ie, a statement of purpose for which the data are required). The data may not be released unless all Bristol Trials Centre and sponsor requirements are fulfilled (eg, protocol, sample size calculation, REC and Health Research Authority approval if appropriate, a collaboration arrangement in place between the sponsor and the external body).

**Author affiliations**
[1]Clinical Trials and Evaluation Unit, University of Bristol Faculty of Medical and Veterinary Sciences, Bristol, UK
[2]Centre for Surgical Research, School of Social and Community Medicine, University of Bristol, Bristol, UK
[3]Trent Oesophago-Gastric Unit, Nottingham City Hospital, Nottingham, UK
[4]Bariatric Unit, Department of Surgery, Sunderland Royal Hospital, Sunderland, UK
[5]School of Population Health Sciences, University of Bristol, Bristol, UK
[6]Clinical Radiology, Leeds Teaching Hospitals NHS Trust, Leeds, UK
[7]NHS Coventry and Rugby Clinical Commissioning Group, Coventry, UK
[8]Centre for Surgical Research, Population Health Sciences, University of Bristol, Bristol, UK
[9]Division of Gastrointestinal Surgery, Nottingham University Hospitals NHS Trust, Nottingham, UK
[10]Department of Upper GI Surgery, University Hospitals Birmingham NHS Foundation Trust, Birmingham, UK
[11]Department of Hepatobiliary and Transplantation Surgery, Leeds Teaching Hospitals NHS Trust, Leeds, UK

**Acknowledgements** The authors would like to thank all study staff involved in recruitment, coordination and data collection for this study, and members of the patient and public involvement group who assisted with study design and review of study documents. Special thanks are also given to the clinical and nursing teams at participating centres for their support in the conduct of the study. The authors would like to acknowledge the data monitoring and safety committee (DMSC) and study steering committee (SSC) for their oversight of the study. The independent DMSC is formed of a chairperson, one consultant surgeon and one consultant radiologist. The SSC is formed of a chairperson, two consultant surgeons, two consultant radiologists, one medical statistician and two public and patient involvement representatives; other SSC members with observer status represent the study executive group (SEG) in these committees. The Sunflower SEG is made up of the following: Madeleine Clout, Jane Blazeby, Chris Rogers, Barnaby Reeves, Michelle Lazaroo, Kerry Avery, Natalie Blencowe, Ravi Vohra, Neil Jennings, William Hollingworth, Joanna Thorn, Marcus Jepson, Jane Collingwood, Ashley Guthrie, Elizabeth Booth, Samir Pathak, Ian Beckingham, Lucy Culliford, Ewen Griffiths, Raneem Albazaz and Giles Toogood. Patients and/or the public were involved in the design of this research. Patients and/or the public are/will be involved in the conduct, reporting and dissemination plans of this research. Refer to the Methods section for further details. The study is formally supported by the Association of Upper Gastrointestinal Surgeons of Great Britain and Ireland and the Great Britain & Ireland Hepato Pancreato Biliary Association, both of which recognise the importance of the objectives of this study.

**Contributors** MC: preparation and drafting of study protocol, first draft and writing of manuscript. JB: study concept and design, preparation and review of study protocol, prestudy feasibility work, definition of study intervention and outcomes, review of manuscript and support for Chief Investigator. CR: study design, preparation and drafting of study protocol, sample size and statistical analysis plan, and review of manuscript. BR: study concept and design, preparation and drafting of study protocol, and review of manuscript. ML: statistical analyses, statistical analysis plan and review of manuscript. KA: study design, review of study protocol, leading patient and public involvement, and review of manuscript. NB: study concept and design, review of study protocol and review of manuscript. RV: study concept and design, review of study protocol and review of manuscript. NJ: study concept and design, review of study protocol and review of manuscript. WH: study design, review of study protocol, health economic analysis plan and analyses, and review of manuscript. JT: health economic analysis plan and analyses, and review of manuscript. MJ: study design, review of study protocol, leading recruitment intervention and review of manuscript. JC: review of manuscript. AG: study design, review of study protocol and review of manuscript. EB: study design, review of study protocol and review of manuscript. SP: study design, review of study protocol and review of manuscript. IB: study concept and design, review of study protocol and review of manuscript. LC: study design, review of study protocol and review of manuscript. EG: study concept and design, review of study protocol and review of manuscript. RA: study concept and design, review of study protocol and review of manuscript. GT: study concept and design, preparation of study protocol, definition of study intervention and outcomes, review of manuscript, and study Chief investigator. All authors read and approved the final manuscript.

**Funding** This work is supported by the National Institute for Health Research (NIHR) Health Technology Assessment Programme (Grant Ref: 16/142/04). The study is also supported by the Royal College of Surgeons Surgical Trials Centre in Bristol. JB, CR, BR and KA are all supported by the NIHR Bristol and Weston Biomedical Research Centre. JB is an NIHR senior investigator. The study sponsor is Leeds Teaching Hospitals NHS Trust (Research Governance Manager Anne Gowing, anne.gowing@nhs.net), and the study is managed by the Bristol Trials Centre, Clinical Trials and Evaluation Unit. This study was designed and is being delivered in collaboration with the Bristol Trials Centre, Clinical Trials and Evaluation Unit, a UK Clinical Research Collaboration registered clinical trials unit, which is in receipt of NIHR clinical trials unit support funding. The research team acknowledges the support of the NIHR Clinical Research Network. The Sunflower Study is overseen by an independent SSC and DMSC. The study has been designed with input from public and patient groups.

**Disclaimer** The funder had no role in the design of the study, data collection or writing this manuscript.

**Competing interests** None declared.

**Patient consent for publication** Not required.

**Provenance and peer review** Not commissioned; externally peer reviewed.

peer-reviewed. Any opinions or recommendations discussed are solely those of the author(s) and are not endorsed by BMJ. BMJ disclaims all liability and responsibility arising from any reliance placed on the content. Where the content includes any translated material, BMJ does not warrant the accuracy and reliability of the translations (including but not limited to local regulations, clinical guidelines, terminology, drug names and drug dosages), and is not responsible for any error and/or omissions arising from translation and adaptation or otherwise.

**ORCID iDs**
Madeleine Clout http://orcid.org/0000-0001-7645-1199
Jane Blazeby http://orcid.org/0000-0002-3354-3330
Barnaby Reeves http://orcid.org/0000-0002-5101-9487
Kerry Avery http://orcid.org/0000-0001-5477-2418
Natalie S Blencowe http://orcid.org/0000-0002-6111-2175
Joanna Thorn http://orcid.org/0000-0001-8962-2428

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
