## [Reviewer comments · BMJ Open]

ARTICLE DETAILS

TITLE (PROVISIONAL)	Study protocol for a randomised controlled trial to establish the clinical and cost effectiveness of expectant management versus pre-operative imaging with magnetic resonance cholangiopancreatography in patients with symptomatic gallbladder disease undergoing laparoscopic cholecystectomy at low or moderate risk of common bile duct stones (The Sunflower Study)
AUTHORS	Clout, Madeleine; Blazeby, Jane; Rogers, Chris; Reeves, Barnaby; Lazaroo, Michelle; Avery, Kerry; Blencowe, Natalie; Vohra, Ravi; Jennings, Neil; Hollingworth, William; Thorn, Joanna; Jepson, Marcus; Collingwood, Jane; Guthrie, Ashley; Booth, Elizabeth; Pathak, Samir; Beckingham, Ian; Culliford, Lucy; Griffiths, Ewen; Albazaz, Raneem; Toogood, Giles

VERSION 1 – REVIEW

REVIEWER	William G. Hawkins MD Chief, Section of Hepatobiliary, Pancreatic, & Gastrointestinal Surgery Neidorff- Family and Robert C. Packman Professor of Surgery Director, Washington University SPORE in Pancreatic Cancer WUSM, Department of Surgery 660 S. Euclid Avenue, Campus Box 8109 St. Louis MO 63110 USA
REVIEW RETURNED	05-Oct-2020

GENERAL COMMENTS	There are two questions under consideration in design of this trial. The first question is patient safety and the second is which approach is most cost-effective. The study team has designed a trial aimed to evaluate two common practices in gallbladder surgery for those with symptoms and low risk or moderate risk of CBD stones. Do we need to evaluate the common bile duct (CBD) before or during a laparoscopic cholecystectomy? This is an important question and for the most part this is a very well thought out and well powered study from which insights will be obtained. There are some minor weaknesses to the study design. For example, there are two important variables not addressed that might have implications on the interpretation of the study results and which might also affect patient outcomes as well as relative costs. The first variable that might affect this studies interpretation is the potential for differential timelines. It is not clear if the timeline is significantly different if patients are randomized to undergo MRI. Is the MRI done on presentation or on an elective schedule? How long does it take to obtain and does this affect the interval from diagnosis until definitive treatment? When indicated is ERCP done immediately? How soon after the ERCP is surgery? The second
---

	potential confounder is the uncertainty about the necessity of ERCP or lap CBD exploration in the presence of stones. Many asymptomatic stones will pass and there is considerable treatment variability and reimaging or procedures to intervene are not required. For the purposes of such a trial nonaction on these findings makes little sense. Yet non action on a clinical test may occur depending on surgeon preference. The details of the planned cost analysis are not provided. However, I would predict that a few patients in such a situation may have very large costs. For example, just a few cases of infected necrotizing pancreatitis could skew a cost analysis due to their relatively large relative costs. Perhaps these rare catastrophic events might be counted as events and excluded from analysis. Perhaps data from the first phase of the study might be used to inform the plans for the second phase. In summary, this is a large undertaking comparing two acceptable standards and due to the large number of patients involved is sure to be informative.
--	---

REVIEWER	Akiko Kowada Kitasato University Graduate School of Medical Sciences, Japan
REVIEW RETURNED	24-Nov-2020

GENERAL COMMENTS	Reviewer's comments to the authors General comments Common bile duct (CBD) stones are estimated to be present in 10-20% of individuals with symptomatic gallstones. Further investigation of patients at moderate or low risk of CBD stones is not standardised and practice across the United Kingdom (UK) varies. Fewer than 10% of these patients actually have CBD stones. In updated guideline on the management of common bile duct stones by the British Society of Gastroenterology, MRCP or EUS is recommended as a next step unless the patient is proceeding directly to cholecystectomy supplemented by intraoperative cholangiography or laparoscopic ultrasound for patients with an intermediate probability of stones. The aim of the authors was to compare the effectiveness and cost-effectiveness of pre-operative imaging with MRCP versus expectant management in patients with symptomatic gallbladder disease undergoing laparoscopic cholecystectomy who are at low or moderate risk of CBD stones. MRCP produces detailed images of the biliary and pancreatic ducts, involves a 1-hour hospital visit and costs the National Health Service about £365. The authors considered that research is needed to establish whether it is appropriate for patients at low and moderate risk of CBD stones to proceed straight to LC without imaging the CBD with MRCP. In this protocol, participants, clinicians and other hospital staff caring for participants will be unblinded because attempts to blind participants by undertaking "dummy" MRCP in those randomised to EM would have added significant additional research costs and created logistical issues and "dummy" MRCP would cause a similar delay to LC as actual MRCP. This study protocol is well written and well-planned.
--

REVIEWER	Akiko Kowada Kitasato University Graduate School of Medical Sciences, Japan
REVIEW RETURNED	24-Nov-2020

GENERAL COMMENTS	Reviewer's comments to the authors General comments Common bile duct (CBD) stones are estimated to be present in 10-20% of individuals with symptomatic gallstones. Further investigation of patients at moderate or low risk of CBD stones is not standardised and practice across the United Kingdom (UK) varies. Fewer than 10% of these patients actually have CBD stones. In updated guideline on the management of common bile duct stones by the British Society of Gastroenterology, MRCP or EUS is recommended as a next step unless the patient is proceeding directly to cholecystectomy supplemented by intraoperative cholangiography or laparoscopic ultrasound for patients with an intermediate probability of stones. The aim of the authors was to compare the effectiveness and cost-effectiveness of pre-operative imaging with MRCP versus expectant management in patients with symptomatic gallbladder disease undergoing laparoscopic cholecystectomy who are at low or moderate risk of CBD stones. MRCP produces detailed images of the biliary and pancreatic ducts, involves a 1-hour hospital visit and costs the National Health Service about £365. The authors considered that research is needed to establish whether it is appropriate for patients at low and moderate risk of CBD stones to proceed straight to LC without imaging the CBD with MRCP. In this protocol, participants, clinicians and other hospital staff caring for participants will be unblinded because attempts to blind participants by undertaking "dummy" MRCP in those randomised to EM would have added significant additional research costs and created logistical issues and "dummy" MRCP would cause a similar delay to LC as actual MRCP. This study protocol is well written and well-planned.
--

VERSION 1 – AUTHOR RESPONSE

Reviewer 1:

We thank this reviewer for their two comments, which are responded to below.

1.The first important timelines. It is not clear if the timeline is significantly different if patients are randomized to undergo MRI. Is the MRI done on presentation or on an elective schedule? How long does it take to obtain and does this affect the interval from diagnosis until definitive treatment? When indicated is ERCP done immediately? How soon after the ERCP is surgery?

Thank you for the comment. It is possible for the timelines to differ between the two treatment strategies. Sunflower is testing a strategy of looking for CBD stones with an MRI or managing patients expectantly. Differences in timeline is something that we are monitoring closely and is of particular interest in this study. There are no protocolised timelines for the MRI (if randomised to MRI in either the elective or the emergency setting), or an ERCP or lap CBD if the local clinician decides to do this based on the results of the MRI. We intentionally chose a pragmatic design to reflect the pathways of care in the NHS, increasing the generalisability of study findings. There is enormous variation in practice in the UK and to establish evidence relevant to patients, surgeons and health policy makers we chose a design that is relevant to and could be implemented within the complexity of the NHS in the UK. In the study we are collecting the dates on which all these tests and treatments are

conducted, which means that we will be able to study the different impacts of the timelines on outcomes. Most importantly, we will gain data about the effectiveness and cost effectiveness of the strategy of using MRI to look for bile duct stones (or not) in patients undergoing laparoscopic cholecystectomy which will be generalisable across hospitals in the UK. We have edited the manuscript to clarify this point (Lines 275-277).

2. The second potential confounder is the uncertainty about the necessity of ERCP or lap CBD exploration in the presence of stones. Many asymptomatic stones will pass and there is considerable treatment variability and reimaging or procedures to intervene are not required.

It is agreed that many stones will pass spontaneously and do not always need to be removed if identified. The protocol does not stipulate removal of identified CBD stones – this is at the discretion of the local clinician. We have edited the manuscript to provide clarity on this point (Lines 278-280).

Reviewer 2:

We thank this reviewer for these supportive comments.

Reviewer 3:

1. My first query is about the used definition of low or moderate risk of CBD stones: Despite the absence of clear consensus about the definition of preoperative risk of CBD stones, authors have proposed several biological and ultrasound criteria. But, these criteria might lack sensitivity. Several scores (ASGE criteria, Lacaine and Huguier CBD lithiasis predictive score, Grande et al. model) have been reported in the literature. Despite, moderate accuracy to predict CBD stones, adding such scores or algorithms might refine the enrollment criteria.

We agree that clear consensus about the definition of preoperative risk of CBD stones is absent from the literature. The information available has been used to inform our choice for this study (recognising that none are perfect). A secondary aim of this study is to use our data to examine how to classify patients at low and moderate risk of CBD stones. It is anticipated that the Sunflower study will make an important contribution to the evidence because of its large sample size and prospective data collection.

2. The second and main question is about the primary outcome: In fact, this study is designed to compare the clinical benefit of two different strategies before LC for patients with low or moderate risk of CBD stones. First strategy is routine preoperative MRCP. Second strategy is expectant management. Thus, the difference in CBD stones detection between the 2 strategies might impact the therapeutic decision and the clinical course of the patient. The first item (the admission rate for a complication of an untreated CBD stones) defined by the authors to evaluate the primary outcome is relevant. The 2 other criteria for the primary outcome might be considered beyond the objectives of the study and thus alter the quality of results. In fact, no clear correlation will be yielded between choosing MRCP or expectant management and the occurrence of a postoperative complication of LC: Postoperative bile leak might be related to a bile duct injury and thus has no relationship with preoperative diagnostic strategy of CBD stones. Same issue with the third criteria (complications of ERCP). In fact, primary outcome might focus on the rate of misdiagnosed CBD stones and their subsequent direct consequences in terms of hospitalization (first criteria), specific postoperative complications (post-LC jaundice due to CBD stone, biliary fistula directly related to the presence of a CBD stone and hyperpressure...) and secondary ERCP rate. Redefinition of both inclusion criteria and more specific items for the primary outcome are mandatory to improve patient selection and statistical power of the results.

We thank the reviewer for raising these issues. Our study team has spent a long time considering all of them and sought advice and input from our independent oversight committee members. The study also underwent extensive peer review at the funding stage. The reason for including post-operative complications of LC and complications of ERCP is that there is some evidence of increased biliary leakage in patients with retained stones post LC (which may be relevant, especially in the expectant management group). Regarding the complications of ERCP, we consider this especially important as it is possible that patients are over treated with MRI and then ERCP and suffer avoidable complications. These important components of the primary outcome will help inform the study results and interpretation in practice. As well as reporting the primary outcome overall, the individual components will also be reported providing full details of the events that contribute to the primary outcome in each group. We have edited the manuscript to explain and clarify these issues (Lines 491-499).

Additional requests 28/01/2021:

Many thanks for highlighting these additional issues.

We can confirm that we have changed the spelling of the name of co-author Ravi Vohra to match the spelling in the ScholarOne system.

We can confirm that author Jane Blazeby and the initials JMB in the Contributorship statement are for the same author. We have amended the initials in the Contributorship statement to JB.

Our thanks again to the Editorial Office and the three reviewers for the time taken to review this manuscript. We look forward to hearing from you following this revised submission.

Kind regards

Madeleine Clout on behalf of the Sunflower Study Group

VERSION 2 – REVIEW

REVIEWER	Hawkins, William Washington University in Saint Louis School of Medicine
REVIEW RETURNED	06-Feb-2021

GENERAL COMMENTS	The study seeks to answer a clinically relevant question about an observed practice variability in the management of biliary disease. The study is well designed to answer the questions it seeks to answer. The two different practice patterns have different risks for the patients and different costs for the health care system. One major obstacle to completion of this study is that the important clinical events such as severe pancreatitis from a missed CBD stone or severe pancreatitis from an unnecessary ERCP are quite rare. As a result the study design required to sort out the relative risks and benefits of the two practice patterns requires more than 13,000 patients. The investigative team has made some design compromises for practical reasons. For example, the randomization seeks to emulate the current MRI usage with a 2:1 allocation. There is inadequate statistical justification for this protocol decision. It seems to be applied to emulate current costs to the health care system. Can we get to the answer more quickly if we applied a 1:1 randomization? Another example of how the practical approach may slow the study is not to prescribe behavior (treatment) to a clinical finding. The best example of this non-sequitur is the patient who is randomized to receive an MRI scan and is found to have a small stone in the hands of a clinician who decides not to act on this information. In the real world a clinician
--

	who orders this test would act upon the result because there can be no other justification for getting an expensive test that does not direct a therapeutic decision. If we prescribe behavior to MRI findings can we get to the answer with fewer patients? The proposed study has value to clinicians, patients, and to the health care system. The proposed study is going to be the most convincing way to get the answer but there are alternatives in a nationalized health care system. Consensus guidelines can be made that influence clinicians and payers as exemplified by the NCCN guidelines for cancer. In the nationalized health care system it should be possible to implement a practice pattern and compare year over year results. Overall, I think this study has merit and that it will meet its objective and influence practice. Minor issues—Page 15 (lines 283-285)- The event described is an event because it is driven by a clinical symptom and not a cross-over between groups.
--	---

REVIEWER	Rhaiem, Rami Robert Debré University-Hospital
REVIEW RETURNED	17-Feb-2021

GENERAL COMMENTS	I would like to thank the authors for the quality of their work. They answered adequately all the questions I asked after my first revision.
--

VERSION 2 – AUTHOR RESPONSE

Reviewer 1:

We thank this reviewer for their comments, which are responded to below.

1. One major obstacle to completion of this study is that the important clinical events such as severe pancreatitis from a missed CBD stone or severe pancreatitis from an unnecessary ERCP are quite rare. As a result the study design required to sort out the relative risks and benefits of the two practice patterns requires more than 13,000 patients. The investigative team has made some design compromises for practical reasons. For example, the randomization seeks to emulate the current MRI usage with a 2:1 allocation. There is inadequate statistical justification for this protocol decision. It seems to be applied to emulate current costs to the health care system. Can we get to the answer more quickly if we applied a 1:1 randomization?

The reviewer is correct that a 1:1 randomisation ratio would have generated an answer quicker, if this had been feasible. However, as the reviewer surmises, it was important to approximate the current MRI usage for two reasons. First, radiology departments do not have excess capacity and approximating current usage ensures prompt access to MRCP in the imaging group. Second (and related), this ratio minimises any extra treatment costs for MRCP which are paid in the UK by participating hospitals (up to a limit), avoiding the disincentive of additional cost of participation to hospitals. The 1:2 ratio increases the overall size of the trial by about 12% (with only a small increase in trial duration and research cost) but reduces the proportion requiring MRCP by 33%.

2. Another example of how the practical approach may slow the study is not to prescribe behavior (treatment) to a clinical finding. The best example of this non-sequitur is the patient who is randomized to receive an MRI scan and is found to have a small stone in the hands of a clinician who decides not to act on this information. In the real world a clinician who orders this test would act upon the result because there can be no other justification for getting an expensive test that does not direct a therapeutic decision. If we prescribe behavior to MRI findings can we get to the answer with fewer patients?

Although it is interesting to think about being more prescriptive about the treatment of CBD stones found on MRI, it was decided not to do this because different centres do treat them differently (some undertake ERCP, some utilise intraoperative cholangiogram and some choose to observe). The focus of this study is the investigations to look for CBD stones, not how they should be treated. The study does not have concerns relating to the sample size, and over 2,000 patients were recruited in the first 12 months.

3. In response to the minor issue on Page 15 (lines 283-285) we have amended the text in the manuscript from “imaging” to “MRCP”. If participants are allocated to EM and MRCP is carried out, this would constitute a crossover.

Our thanks again to the Editorial Office and the reviewers for the time taken to review this manuscript. We look forward to hearing from you following this revised submission.